

# Genome-wide identification and expression reveal the involvement of the FCS-like zinc finger (FLZ) gene family in *Gossypium hirsutum* at low temperature

JunDuo Wang[1,*], Zhiqiang Li[2,*], Yajun Liang[1], Juyun Zheng[1], Zhaolong Gong[1], Guohui Zhou[2], Yuhui Xu[2] and Xueyuan Li[1]

[1] Cash Crops Research Institute, Xinjiang Academy of Agricultural Science, Urumqi, Xinjiang, China
[2] Adsen Biotechnology Co., Ltd., Urumqi, Xinjiang, China
* These authors contributed equally to this work.

Corresponding authors
Yuhui Xu, genetics_2010@163.com
Xueyuan Li, xjmh2338@163.com

## ABSTRACT

FCS-like zinc finger (FLZ) is a plant-specific gene family that plays an important regulatory role in plant growth and development and its response to stress. However, studies on the characteristics and functions of cotton FLZ family genes are still lacking. This study systematically identified members of the cotton FLZ gene family based on cotton genome data. The cotton FLZ family genes were systematically analyzed by bioinformatics, and their expression patterns in different tissues and under low-temperature stress were analyzed by transcriptome and qRT–PCR. The *G. hirsutum* genome contains 56 FLZ genes distributed on 20 chromosomes, and most of them are located in the nucleus. According to the number and evolution analysis of FLZ family genes, FLZ family genes can be divided into five subgroups in cotton. The *G. hirsutum* FLZ gene has a wide range of tissue expression types, among which the expression is generally higher in roots, stems, leaves, receptacles and calyx. Through promoter analysis, it was found that it contained the most cis-acting elements related to methyl jasmonate (MeJA) and abscisic acid (ABA). Combined with the promoter and qRT–PCR results, it was speculated that *GhFLZ11*, *GhFLZ25*, *GhFLZ44* and *GhFLZ55* were involved in the response of cotton to low-temperature stress. Taken together, our findings suggest an important role for the FLZ gene family in the cotton response to cold stress. This study provides an important theoretical basis for further research on the function of the FLZ gene family and the molecular mechanism of the cotton response to low temperature.

## INTRODUCTION

Cotton is one of the most economically important crops in the world and occupies a very important position in economic development (*Wang et al., 2022*). Chilling damage is a global natural disaster, and it is the main adverse factor affecting the growth and development, geographical distribution, yield and quality of cold-sensitive crops

(*Zhang et al., 2022*). Chilling damage often occurs in various stages of cotton growth and development, causing huge threats and losses to agricultural production (*Zhu et al., 2022*). Cold stress includes chilling stress (0–15 °C) and freezing stress (<0 °C), both of which can greatly affect plant growth and performance (*Thomashow, 1999*; *Yadav, 2010*). Rice, corn, cotton and other important foods and economic crops originate from the tropics and subtropics and thus lack the cold domestication mechanism and have poor adaptability to low temperature. Growth is stunted below 12 °C and temperatures below 0 °C are lethal (*Yang et al., 2015*). Each year, due to low temperatures, cotton production in China, India, the United States and Pakistan has been reduced to varying degrees (*Zhu et al., 2022*). In cotton, low temperature and chilling damage not only cause rotten seeds, rotten buds, rotten roots and insufficient development of cotton bolls but also hinder the transport of photosynthetic products and mineral nutrients to the growing organs, shorten the growth period of cotton, and reduce the yield component factors resulting in decreased cotton yield and quality (*Sanchez, Mangat & Angeles-Shim, 2019*; *Li et al., 2019*). Continuous low temperature stress also induces the occurrence of cotton seedling diseases. In the world's major cotton regions, some cotton fields suffer from low temperature damage to varying degrees every year, which makes cotton germination and seedling growth difficult, cause a lack of cotton seedlings and ridges, and affects cotton growth and development. This poses a huge threat to the cotton industry (*Kaur Dhaliwal et al., 2021*; *Wang et al., 2019*).

The C2-C2 (FCS)-like zinc finger (FLZ) protein is a plant-specific regulatory protein containing an FLZ domain or DUF581 (Domain of Unknown Function 581) (*Jamsheer et al., 2018b*). Some reports suggest that the FLZ protein acts as a fast-folding protein of SnRK1 by interacting with the SnRK1 kinase subunit (*Jamsheer et al., 2018a*). The *MARD1* gene of the Arabidopsis DUF581 gene family is involved in ABA-mediated seed dormancy and can be induced by senescence (*He et al., 2001*). Overexpression of *TaSRHP*, a member of the wheat DUF581 gene family, in *Arabidopsis* can improve plant salt tolerance (*Hou et al., 2013*). In *Arabidopsis*, *AtFLZ6/10* interacts with SnRK1α and inhibits the SnRK1 signaling pathway by inhibiting the accumulation of SnRK1α protein. Consistent with this, *fz6* and *fz10* knockout mutants replicated the phenotype of SnRK1α-overexpressing plants with higher SnRK1α protein levels and growth retardation (*Jamsheer et al., 2018a*; *Jamsheer et al., 2019*). The central role of SnRK1 in regulating plant stress responses has been established, and there is a strong interaction between the FLZ protein and SnRK1 in *Arabidopsis* and maize (*Jamsheer et al., 2018b*; *Jamsheer & Laxmi, 2015*). Gene expression analysis showed that the AtFLZ gene was significantly regulated by environmental cues, such as sugar, ABA, hypoxia, and light, which could positively affect the activity of SnRK1 (*Jamsheer et al., 2019*; *Jamsheer & Laxmi, 2015*; *Börnke, 2014*; *Nietzsche et al., 2016*). Through a Yeast Two-Hybrid Assay (Y2H) assay, eight representative OsFLZ proteins were found to interact strongly with SnRK1A, and *OsFLZ18* was found to regulate the role of early seedling growth by interacting with *SnRK1A* (*Laxmi, 2014*). Expression analysis found that AtFLZ transcripts accumulated in senescent leaves of *Arabidopsis*, suggesting that the FLZ gene family also plays a role in senescence (*Jamsheer et al., 2018a*; *Hou et al., 2013*). Therefore, the plant FLZ gene family may play an important role in plant growth and development and stress resistance.

FLZ genes are an understudied plant-specific gene family (*Chen et al., 2021*). Earlier studies in model plant species found that they were associated with senescence and ABA-mediated seed dormancy (*He et al., 2001*). They are very small proteins, almost all of which contain only one FLZ domain (*Chen et al., 2021*). The FLZ gene family is a plant-specific gene family. In recent years, the identification and functional studies of plant FLZ family genes have attracted increasing attention, but most of the research results were from studies of *Arabidopsis*. There is no report on the phylogenetic analysis of the FLZ gene family in cotton. Increasing research has indicated that the function of FLZ genes is complex and diverse, but whether they play a role in cotton cold resistance is still unknown. Considering this issue, this study performed genome-wide identification of the cotton FLZ gene family and explored the phylogeny, gene structure, and promoter elements of members of the *G. hirsutum* FLZ gene family through bioinformatics analysis using different tissues and low temperatures. Under low temperature treatment, transcriptome and qRT–PCR data were obtained and screened to identify potential GhFLZ functional genes in response to low-temperature stress, and these results provide a reference for studying the function of FLZ genes.

## MATERIALS AND METHODS

### Plant material

We selected 86-1 (cold resistance) and Lumian 2 (cold susceptibility) plant cultivars as the materials for expression analysis under low-temperature stress (*Ge et al., 2021*). The stress temperature used was 12 °C. Seeds with uniform size were selected, and after disinfection with 75% alcohol, the seeds were soaked for 24 h until they turned white. Then, the seeds were sown into 10 cm × 10 cm pots filled with a mixture of vermiculite and sterilized farm soil (in a ratio of 1:2) at room temperature with a relative humidity of 70%. When the cotton plant was at the three-leaf stage, it was put into an artificial climate incubator (RGX-400P; Taisite, New York, NY, USA) at a temperature of 12 °C and a relative humidity of 70% with a 16 h light/8 h dark cycle, and the roots, stems and leaf tissues were collected at 0, 6, 12, 24 and 48 h under stress. Two materials were cryogenically treated with three biological replicates per time point. The samples were quickly frozen with liquid nitrogen and stored in a −80 °C freezer.

### Identification and bioinformatics analysis of the FLZ gene family in cotton

The genomic and proteomic data of *Gossypium arboreum*, *Gossypium raimondii*, *Gossypium hirsutum* and *Gossypium barbadense* were obtained from the COTTONGEN (http://www.cottongen.org/) database. Protein sequences in cotton were identified by performing a hidden Markov model of the FLZ gene domain (PF04570) and a hidden Markov model (HMM) in Hmmsearch software (http://hmmer.org/) (*Potter et al., 2018*). Domain identification was performed on the GhFLZ protein sequence through the NCBI CDD database (*Lu et al., 2020*). Finally, ExPASy and PSORT were used to analyze protein sequence physical parameters and subcellular localization (*Chang et al., 2013*).

## Phylogenetic and collinear analysis of the cotton FLZ gene family

Using the default settings of Clustal W in MEGA 8 software, the FLZ protein sequences of *Arabidopsis thaliana*, rice, *Gossypium arboreum*, *Gossypium raimondii*, *Gossypium hirsutum* and *Gossypium barbadense* were used for multiple sequence alignment. Based on the results of the sequence alignment, the neighbor-joining method was used to build a phylogenetic tree with the bootstrap value set to 1,000 (*Kumar, Stecher & Tamura, 2016*). The resulting phylogenetic tree was beautified with the online tool Evolview3 (https://evolgenius.info/).

## Chromosomal location, gene structure and motif analysis of the FLZ gene family in *Gossypium hirsutum*

The chromosomal location information of FLZ gene family members was extracted from the *Gossypium hirsutum* genome annotation file, and the chromosomal location map of the FLZ genes was drawn by Mapchart software. The evolutionary tree was constructed for the FLZ gene family in *Gossypium hirsutum* by MEGA8 software, and the nwk file was obtained. Motif analysis was conducted by the MEME program (number of functional domains set to 10, minimum width set to 60, and maximum width set to 100) (*Bailey et al., 2009*). The XML file, the NWK file of the evolutionary tree and the GFF file of the gene structure were processed and visualized by TBtools software (*Chen et al., 2020a*).

## Analysis of upstream *cis*-acting elements of the FLZ gene family members in *Gossypium hirsutum*

The 2,000 bp DNA sequence upstream of the FLZ genes in *G. hirsutum* was intercepted, and the possible *cis*-acting elements were predicted using the PlantCARE database (http://bioinformatics.psb.ugent.be/webtools/plantcare/html/) and visualized using the R language ggplot package (*Lescot et al., 2002*).

## RNA-seq analysis

Transcriptome data of organs (root, stem, leaf, pistil, stamen, calyx, petal and receptacle) in *Gossypium hirsutum* under low-temperature stress were downloaded from the NCBI SRA (Sequence Read Archive) database (Genome sequencing project accession: PRJNA248163) (*Zhang et al., 2015*). Data filtering (*i.e.*, exclusion of reads of too low quality and reads that were too short, cutting of adapters, and trimming of polyX tails in 3′ ends to remove unwanted polyX tailing) and quality control were performed with fastp (https://github.com/OpenGene/fastp) software, and the resulting clean data were used for subsequent analysis (*Chen et al., 2018b*). The *Gossypium hirsutum* (TM-1) genome was used as a reference for read alignment (http://www.cottongen.org/), and String Tie was applied to quantify the aligned reads (*Kovaka et al., 2019*; *Kim et al., 2019*; *Ramirez-Gonzalez et al., 2012*; *Liao, Smyth & Shi, 2014*). FPKM (fragments per kilobase of exon per million fragments mapped) refers to the number of reads per kilobase mapped to an exon in every million reads on a map, and the FPKM method was used to assess gene expression. The expression heatmap was drawn with the R language pheatmap package (*Kolde, 2019*).

## qRT–PCR analysis

According to the cDNA information of *G. hirsutum*, 5′ and 3′ primers were designed at the specific region of the gene sequence using Primer 5.0 software (Table S1). Root tissue cDNA was used as the template, and the expression of candidate genes was measured by qRT–PCR. Each sample was tested three times, and the internal reference gene was *GhUBQ7*. qRT–PCR was performed as previously reported (*Zhao et al., 2021*). A total RNA extraction kit (Tiangen, Sichuan, China) was applied. Reverse transcription was performed using an M-MLV RTase cDNA Synthesis Kit (TaKaRa, Kusatsu, Japan). qRT–PCR was performed using the Roche LightCycler® 480II System under the following conditions: 95 °C 15 s, followed by 40 cycles of 95 °C for 15 s, 55 °C for 15 s, and 72 °C for 15 s. The relative quantitative analysis was performed using the $2^{-\Delta\Delta Ct}$ method.

# RESULTS

## Identification of the cotton FLZ gene family

To systematically study the copy number changes of the FLZ gene family during cotton evolution, using the protein domain PF04570, a total of 168 proteins encoding FLZ were identified in four cotton genomes, and the search results were verified in the NCBI-CDD database (Fig. S1). *G. arboreum*, *G. raimondii*, *G. hirsutum* and *G. barbadense* were found to have 29, 28, 56 and 55 FLZ proteins, respectively. This result indicates that in the process of chromosome doubling and cotton evolution, the FLZ gene family does not exist as a result of gene loss or chromosomal rearrangement. We named the *G. hirsutum* FLZ proteins GhFLZ1–GhFLZ56 according to the chromosomal locations of the 56 proteins. The length of the open reading frame (ORF) of the *G. hirsutum* FLZ gene family members ranges from 270 to 1,215 bp, the encoded proteins contain 89–404 amino acid residues, and the differences between the different FLZ proteins are small. The relative molecular weights ranged from 10.18 to 44.64 kDa, and the theoretical isoelectric points ranged from 4.74 to 10.72, indicating that the physicochemical properties of the FLZ gene family members were not very different. The subcellular localization of the proteins showed that 46 were localized to the nucleus and 10 were localized to the chloroplast (Table 1).

To investigate the genomic distribution of *G. hirsutum* FLZ genes on chromosomes, we investigated the chromosomal location of GhFLZ. Fifty-five GhFLZ genes were distributed on 20 chromosomes of *G. hirsutum*, and one gene that could not be clearly mapped to any chromosome was named *GhFLZ56* (Fig. 1). Subgroup A contained 27 FLZ genes, and subgroup D contained 27 FLZ genes. Previous studies suggested that *G. arboreum* and Redmond cotton were the donor species of the *G. hirsutum* A subgenome and D subgenome, respectively, and the number of FLZ genes in *G. hirsutum* subgroup A was two less than that in G. *arboreum* subgroup D. The number of FLZ genes in the subgroup was consistent with the number of FLZ genes in Redmond cotton. This indicates that the FLZ gene may have been conserved during the evolution of *G. hirsutum*, and tandem duplication and that segmental duplication events did not occur. In *G. hirsutum*, there are three sequences of the FLZ gene family members on chromosome A02 but only one sequence on chromosome D02. There are two sequences of the FLZ gene family members

**Table 1 The information of FLZ gene family in *G. hirsutum*.**

| Gene name | Gene ID | Open reading frame/bp | Protein length/ aa | Relative molecular weight /KDa | Theoretical isoelectric point (pI) | Subcellular localization |
|---|---|---|---|---|---|---|
| GH_A02G0517 | GhFLZ01 | 411 | 136 | 15.0704 | 9.91 | Nucleus |
| GH_A02G1569 | GhFLZ02 | 510 | 169 | 19.1513 | 9.6 | Nucleus |
| GH_A02G1570 | GhFLZ03 | 507 | 168 | 18.8131 | 9.85 | Nucleus |
| GH_A03G0080 | GhFLZ04 | 543 | 180 | 20.3144 | 7.98 | Nucleus |
| GH_A03G0309 | GhFLZ05 | 639 | 212 | 23.3111 | 7.2 | Nucleus |
| GH_A05G1293 | GhFLZ06 | 1161 | 386 | 42.605 | 4.74 | Nucleus |
| GH_A05G2289 | GhFLZ07 | 507 | 168 | 18.6861 | 10.09 | Nucleus |
| GH_A06G0408 | GhFLZ08 | 516 | 171 | 18.605 | 10.12 | Nucleus |
| GH_A06G0858 | GhFLZ09 | 663 | 220 | 24.2484 | 8.69 | Nucleus |
| GH_A06G1671 | GhFLZ10 | 474 | 157 | 17.9858 | 10.33 | Nucleus |
| GH_A07G0167 | GhFLZ11 | 681 | 226 | 25.4588 | 9.23 | Nucleus |
| GH_A08G0626 | GhFLZ12 | 276 | 91 | 10.3595 | 8.47 | Nucleus |
| GH_A08G0627 | GhFLZ13 | 270 | 89 | 10.1842 | 7.86 | Nucleus |
| GH_A08G0722 | GhFLZ14 | 750 | 249 | 27.9175 | 8.94 | Nucleus |
| GH_A08G1875 | GhFLZ15 | 519 | 172 | 19.6897 | 7.76 | Nucleus |
| GH_A09G1019 | GhFLZ16 | 1137 | 378 | 42.1565 | 4.91 | Nucleus |
| GH_A09G2141 | GhFLZ17 | 522 | 173 | 19.4145 | 8.65 | Nucleus |
| GH_A09G2379 | GhFLZ18 | 840 | 279 | 31.1418 | 6.14 | Nucleus |
| GH_A09G2389 | GhFLZ19 | 453 | 150 | 16.9906 | 10.66 | Chloroplast |
| GH_A09G2584 | GhFLZ20 | 402 | 133 | 14.7627 | 8.49 | Nucleus |
| GH_A10G0516 | GhFLZ21 | 540 | 179 | 20.099 | 9.4 | Nucleus |
| GH_A10G1806 | GhFLZ22 | 420 | 139 | 15.4326 | 9.82 | Chloroplast |
| GH_A11G0766 | GhFLZ23 | 276 | 91 | 10.5106 | 7.91 | Nucleus |
| GH_A11G1293 | GhFLZ24 | 864 | 287 | 32.1551 | 8.21 | Nucleus |
| GH_A11G2087 | GhFLZ25 | 465 | 154 | 16.7898 | 9.76 | Nucleus |
| GH_A12G0039 | GhFLZ26 | 714 | 237 | 26.4131 | 8.75 | Nucleus |
| GH_A12G2325 | GhFLZ27 | 522 | 173 | 19.3803 | 8.03 | Nucleus |
| GH_D02G0537 | GhFLZ28 | 411 | 136 | 14.9292 | 9.63 | Nucleus |
| GH_D03G0441 | GhFLZ29 | 510 | 169 | 19.1063 | 9.29 | Nucleus |
| GH_D03G1666 | GhFLZ30 | 639 | 212 | 23.332 | 7.2 | Chloroplast |
| GH_D03G1882 | GhFLZ31 | 540 | 179 | 20.1301 | 7.7 | Nucleus |
| GH_D05G1293 | GhFLZ32 | 1215 | 404 | 44.6412 | 5.07 | Nucleus |
| GH_D05G2310 | GhFLZ33 | 507 | 168 | 18.7783 | 10.3 | Nucleus |
| GH_D06G0388 | GhFLZ34 | 516 | 171 | 18.5629 | 10.12 | Nucleus |
| GH_D06G0837 | GhFLZ35 | 663 | 220 | 24.1252 | 8.69 | Nucleus |
| GH_D06G1690 | GhFLZ36 | 474 | 157 | 18.0259 | 10.18 | Nucleus |
| GH_D07G0176 | GhFLZ37 | 681 | 226 | 25.4889 | 9.23 | Nucleus |
| GH_D08G0624 | GhFLZ38 | 276 | 91 | 10.3595 | 8.47 | Nucleus |
| GH_D08G0625 | GhFLZ39 | 276 | 91 | 10.3705 | 8.73 | Nucleus |
| GH_D08G0712 | GhFLZ40 | 750 | 249 | 28.0608 | 8.84 | Nucleus |
| GH_D08G1890 | GhFLZ41 | 519 | 172 | 19.6897 | 7.76 | Nucleus |

| Table 1 (continued) | | | | | | |
|---|---|---|---|---|---|---|
| Gene name | Gene ID | Open reading frame/bp | Protein length/aa | Relative molecular weight /KDa | Theoretical isoelectric point (pI) | Subcellular localization |
| GH_D08G2152 | GhFLZ42 | 348 | 115 | 13.091 | 10.48 | Chloroplast |
| GH_D09G0968 | GhFLZ43 | 1137 | 378 | 42.2956 | 4.85 | Nucleus |
| GH_D09G2076 | GhFLZ44 | 522 | 173 | 19.4145 | 8.65 | Nucleus |
| GH_D09G2317 | GhFLZ45 | 840 | 279 | 30.9545 | 6.14 | Nucleus |
| GH_D09G2327 | GhFLZ46 | 453 | 150 | 17.0347 | 10.72 | Chloroplast |
| GH_D09G2514 | GhFLZ47 | 387 | 128 | 14.3763 | 8.49 | Nucleus |
| GH_D10G0542 | GhFLZ48 | 540 | 179 | 20.2122 | 10.34 | Nucleus |
| GH_D10G1909 | GhFLZ49 | 420 | 139 | 15.4306 | 9.84 | Chloroplast |
| GH_D11G0801 | GhFLZ50 | 276 | 91 | 10.5046 | 8.61 | Chloroplast |
| GH_D11G1320 | GhFLZ51 | 864 | 287 | 32.1001 | 8.35 | Nucleus |
| GH_D11G2345 | GhFLZ52 | 477 | 158 | 17.3584 | 9.45 | Chloroplast |
| GH_D12G0039 | GhFLZ53 | 726 | 241 | 26.8826 | 8.75 | Chloroplast |
| GH_D12G1078 | GhFLZ54 | 450 | 149 | 16.8039 | 8.51 | Nucleus |
| GH_D12G2340 | GhFLZ55 | 522 | 173 | 19.4113 | 7.71 | Nucleus |
| GH_scaffold675-4_objG0001 | GhFLZ56 | 357 | 118 | 13.4314 | 9.94 | Chloroplast |

on A03 and A12, D03 and D12 each have three sequences, and the remaining FLZ gene family member sequences on the chromosomes are distributed in stacks. This indicates that the *G. hirsutum* FLZ gene family may have lost a certain FLZ gene in the process of evolution. However, on the whole, the A subgroup and the D subgroup still have a strong corresponding relationship, which is also in line with the evolutionary relationship of cotton.

### Evolutionary analysis of cotton FLZ genes

To further understand the evolutionary relationship of *G. hirsutum* FLZ, we constructed a phylogenetic tree of the full-length sequences of 56 *G. hirsutum* FLZ proteins, 27 rice FLZ proteins and 16 *Arabidopsis thaliana* FLZ protein (Fig. 2A). According to the phylogenetic tree grouping results of Pseudomonas and rice, FLZ proteins were divided into five groups. Each group has at least two FLZ proteins from the monocotyledonous plants *Arabidopsis thaliana* and rice, indicating that the differentiation time of the FLZ gene family was earlier than that of monocotyledonous plants. Except for Group 3, the number of FLZ proteins of cotton in each subgroup is much greater than that of *Arabidopsis* and rice, which indicates that the FLZ gene family has undergone obvious tandem duplication in the process of evolution, resulting in tetraploid cotton containing more FLZ genes. Group 3 contains only four FLZ proteins of *G. hirsutum* but 7 FLZ proteins of rice, which indicates that different subgroups of FLZ may have different biological functions, and the number of different subgroups is very different. It is speculated that different subgroups are responsible for functions that may vary.

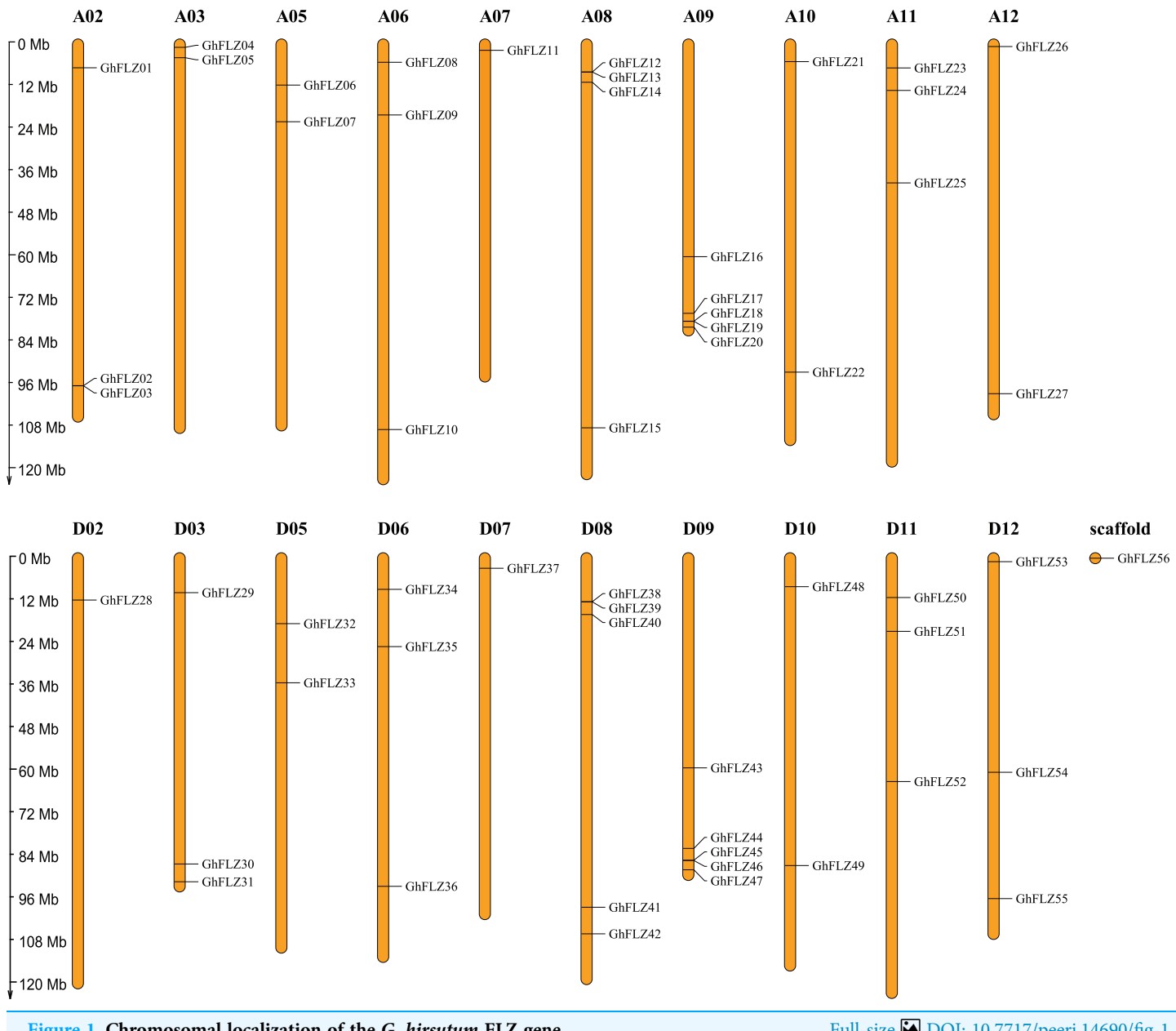

**Figure 1 Chromosomal localization of the *G. hirsutum* FLZ gene.**

To further explore the phylogenetic relationship of the cotton FLZ gene family, an evolutionary tree was constructed using the protein sequences of four different cotton FLZ genes (Fig. 2B). This is consistent with the results of the previous analysis and in line with the evolutionary relationship of cotton. The results showed that the FLZ gene family members are relatively conserved in the evolution of cotton. Although Group 3 members are relatively few, they have always existed in the evolution of cotton, which indicates that they may play an important role in the development of cotton and the process of stress.

To further infer the evolutionary relationship of FLZ genes in *G. hirsutum*, intrachromosomal collinearity analysis of FLZ genes in *G. hirsutum*, rice and *Arabidopsis*

A

B

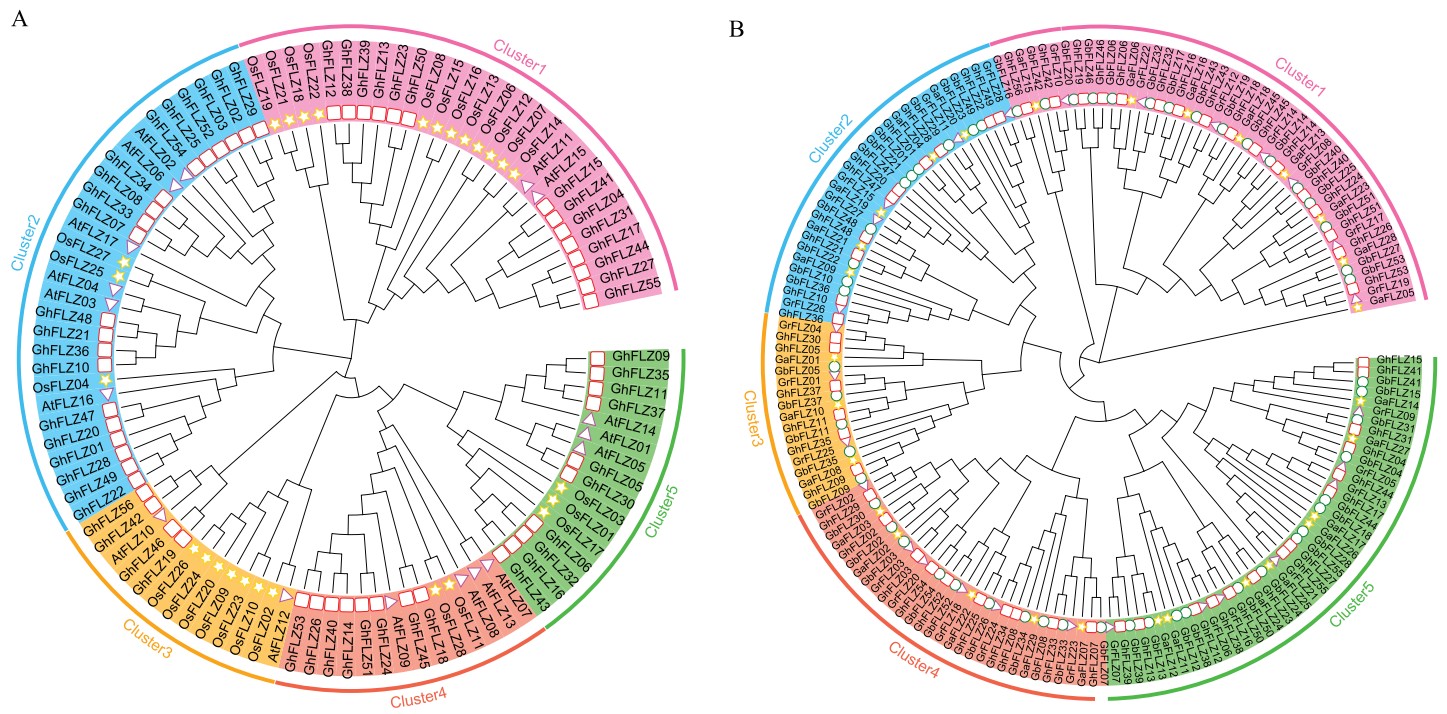

**Figure 2 Phylogenetic tree of the cotton FLZ gene family.** (A) Phylogenetic tree of *Arabidopsis thaliana*, rice and *G. hirsutum*. (B) Phylogenetic tree of *G. arboreum*, *G. raimondii*, *G. barbadense* and *G. hirsutum*.

was performed using Circos software (Fig. 3A). The results showed that there were a large number of collinear blocks among the three plants (Fig. 3A). The collinearity analysis identified six pairs of FLZ genes in *G. hirsutum* and *Arabidopsis thaliana* and 21 pairs of FLZ genes in *G. hirsutum* and rice. In addition, six *G. hirsutum* FLZ genes were homologous to both rice and *Arabidopsis* FLZ genes, indicating that plant FLZ genes may have evolved from a common ancestor of different plants.

According to gene number, chromosomal location and phylogenetic tree analysis, FLZ was found to be conserved in cotton evolution. To deeply study the evolutionary relationship of cotton FLZ, we chose *G. hirsutum* as the core species and constructed the collinear relationships between *G. hirsutum* and *G. arboreum* FLZ proteins and between *G. raimondii* and *G. barbadense* FLZ proteins (Fig. 3B). Among them, *GhFLZ04* and *GhFLZ20* did not have any collinear sequences with any *G. barbadense* FLZ proteins, *GhFLZ19* and *GhFLZ30* did not have any collinear sequences with any *G. arboreum* FLZ proteins, *GhFLZ46* did not have any collinear sequences with any *G. raimondii* FLZ proteins, and *GhFLZ43* did not have any collinear sequences with any *G. raimondii* or *G. barbadense* FLZ proteins. We found that, in addition to these six genes, the FLZ family gene sequences in the A subgenome of *G. hirsutum* were collinear with those of *G. arboreum* and *G. barbadense*, and the FLZ gene sequences in the D subgenome were colinear with those in *G. raimondii* and *G. barbadense*. These results suggest that FLZ was relatively conserved during the evolution of cotton.
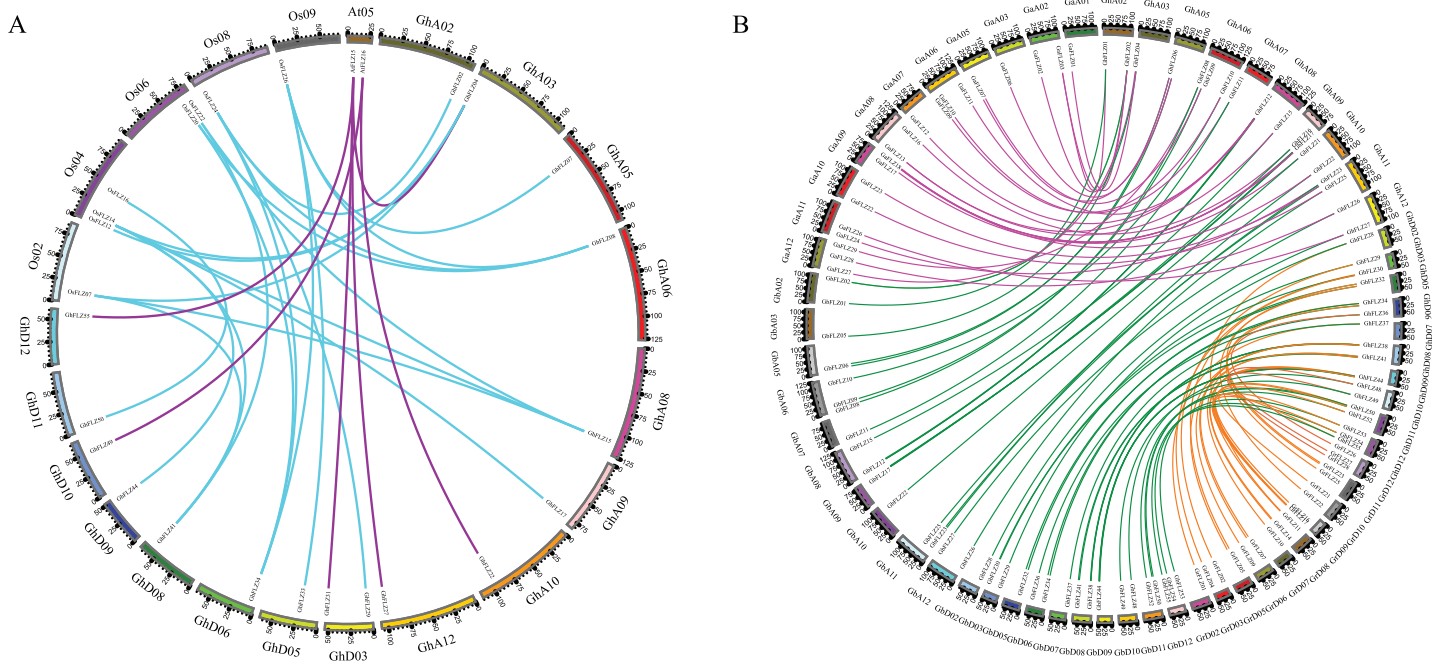

**Figure 3 Chromosome collinearity analysis of cotton FLZ gene.** (A) FLZ collinearity in *G. hirsutum*, rice and *Arabidopsis*. The purple line represents the collinearity of each *Arabidopsis* in *G. hirsutum*, and the blue line represents the collinearity between *G. hirsutum* and rice. (B) FLZ collinearity analysis in *G. arboreum, G. raimondii, G. hirsutum* and *G. barbadense*. Purple lines represent the collinearity of *G. hirsutum* and *G. arboreum* A subgenomic FLZ genes. Green lines represent the collinearity of *G. hirsutum* and *G. barbadense* FLZ genes. The orange line represents the collinearity of *G. hirsutum* and *G. raimondii* D subgenomic FLZ genes.

## Evolutionary tree, gene structure and motif analysis of the FLZ genes of *G. hirsutum*

The phylogenetic tree, gene structure and motif analyses were performed based on the full-length, CDS and protein sequences of the *G. hirsutum* FLZ genes (Fig. 4). *G. hirsutum* FLZ members were divided into five subgroups according to the phylogenetic tree results. All FLZ genes except *GhFLZ32* contain two exons, and all contain an identical motif 1. The Group 1 subgroup contained motifs 3 and 9, the Group 5 subgroup contained motifs 3, 5 and 9, and the other subgroups did not contain any motifs. Although the *GhFLZ32* gene contains 3 CDSs, it has no redundant motifs, which indicates that the structural differences between *GhFLZ32* and genes in the same family may be caused by changes in gene function or errors in genome annotation. In the *G. hirsutum* FLZ gene family, most members of the same subgroup have similar motifs, lengths and structures, suggesting that they are functionally similar. The protein sequences of the same subgroup were highly conserved, but there were significant differences among the groups, especially for the sequences of Group 1 and Group 5.

## Analysis of *cis*-acting elements of the *G. hirsutum* FLZ gene promoter

Transcription factors (TFs) can regulate plant growth and development and responses to stress, including responses to hormones and environmental factors, cell differentiation and organ development, by regulating gene expression. To further study the possible

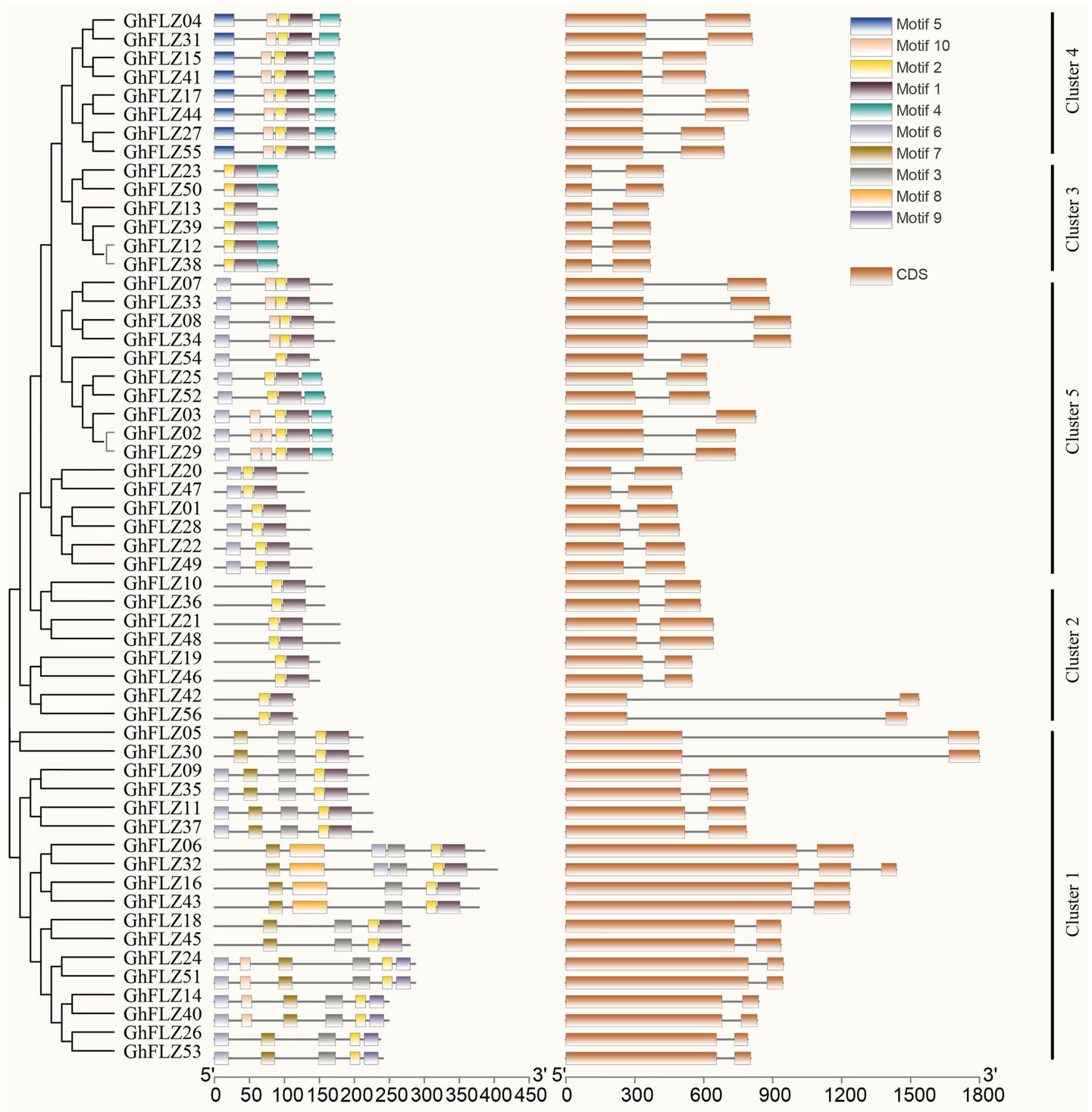

**Figure 4 Evolutionary tree, gene structure and conserved motif analysis of the *G. hirsutum* FLZ gene family.**

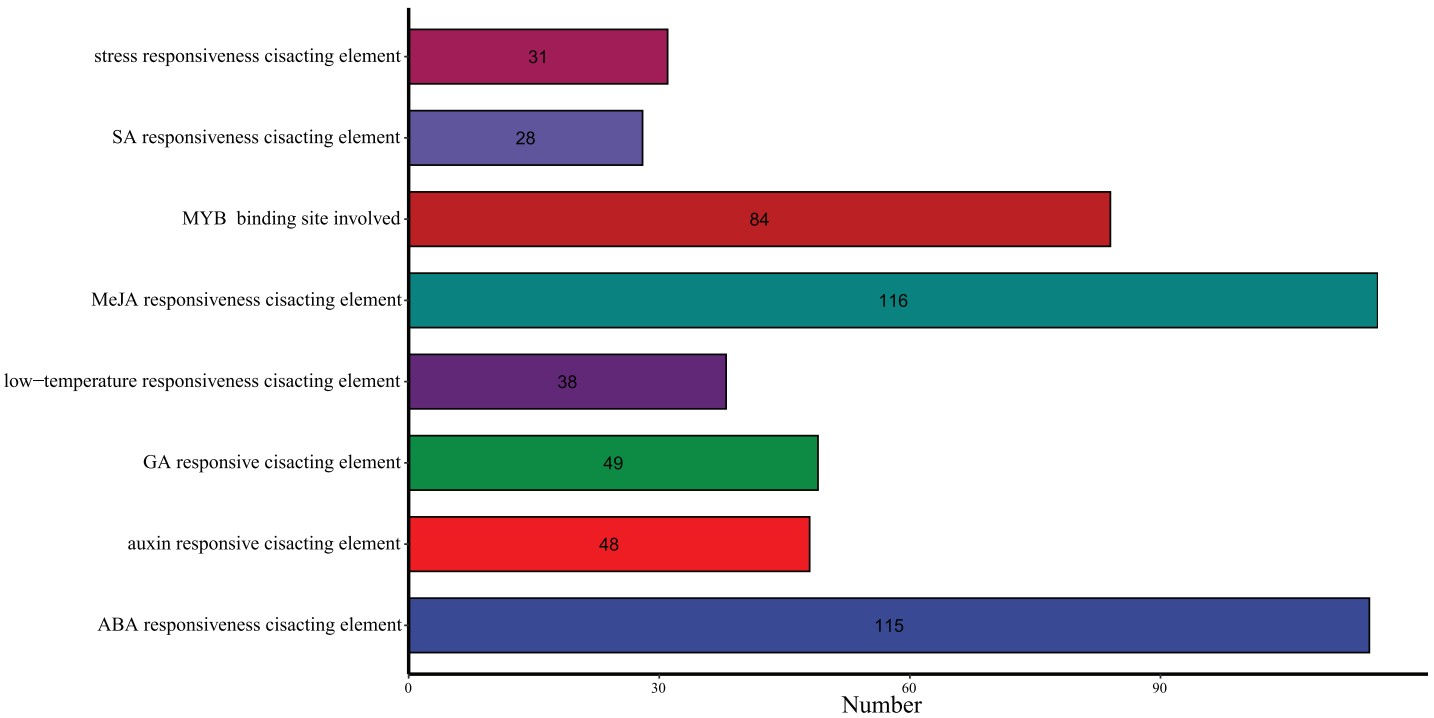

**Figure 5** Quantitative analysis of *cis*-acting elements in the promoter region of *G. hirsutum* FLZ family genes.

transcriptional regulation mechanism of the *G. hirsutum* FLZ gene family, the *cis*-acting elements in the upstream 2,000 bp promoter sequences of 56 GhFLZ genes were analyzed (Fig. 5). The results showed that these *cis*-elements were mainly divided into hormone and stress response elements. The *cis*-acting elements related to various hormones and biotic stress responses in the GhFLZ gene promoter were summarized, and the results showed that the GhFLZ gene contained at least eight related *cis*-elements in its promoter region, including five hormone elements and two stress-related elements. In contrast to the evolution results of GhFLZ, the distributions of these two types of elements in the closely related GhFLZ promoter sequences were not similar (Fig. S2). Each FLZ gene promoter contains different numbers and types of *cis*-acting elements, indicating that they may participate in different biotic and abiotic stress responses through different signaling pathways. Furthermore, of all *cis*-element types in the GhFLZ genes, SA-responsive elements were the least frequent, with most *cis*-acting elements being related to MeJA and ABA, indicating that the cotton FLZ gene family may exert its biological functions mainly through the MeJA and ABA pathways. We speculate that external environmental stress can induce the expression of the GhFLZ gene through its response to *cis*-regulatory elements and further improve the resistance of cotton to environmental stress.

### *G. hirsutum* FLZ gene expression analysis

To elucidate the spatial expression pattern of the *G. hirsutum* FLZ genes, we analyzed the expression pattern of the *G. hirsutum* FLZ genes in eight tissues: cotton root, stem, leaf, receptacle, calyx, petal, stamen and pistil. *G. hirsutum* FLZ genes were all tissue-specifically

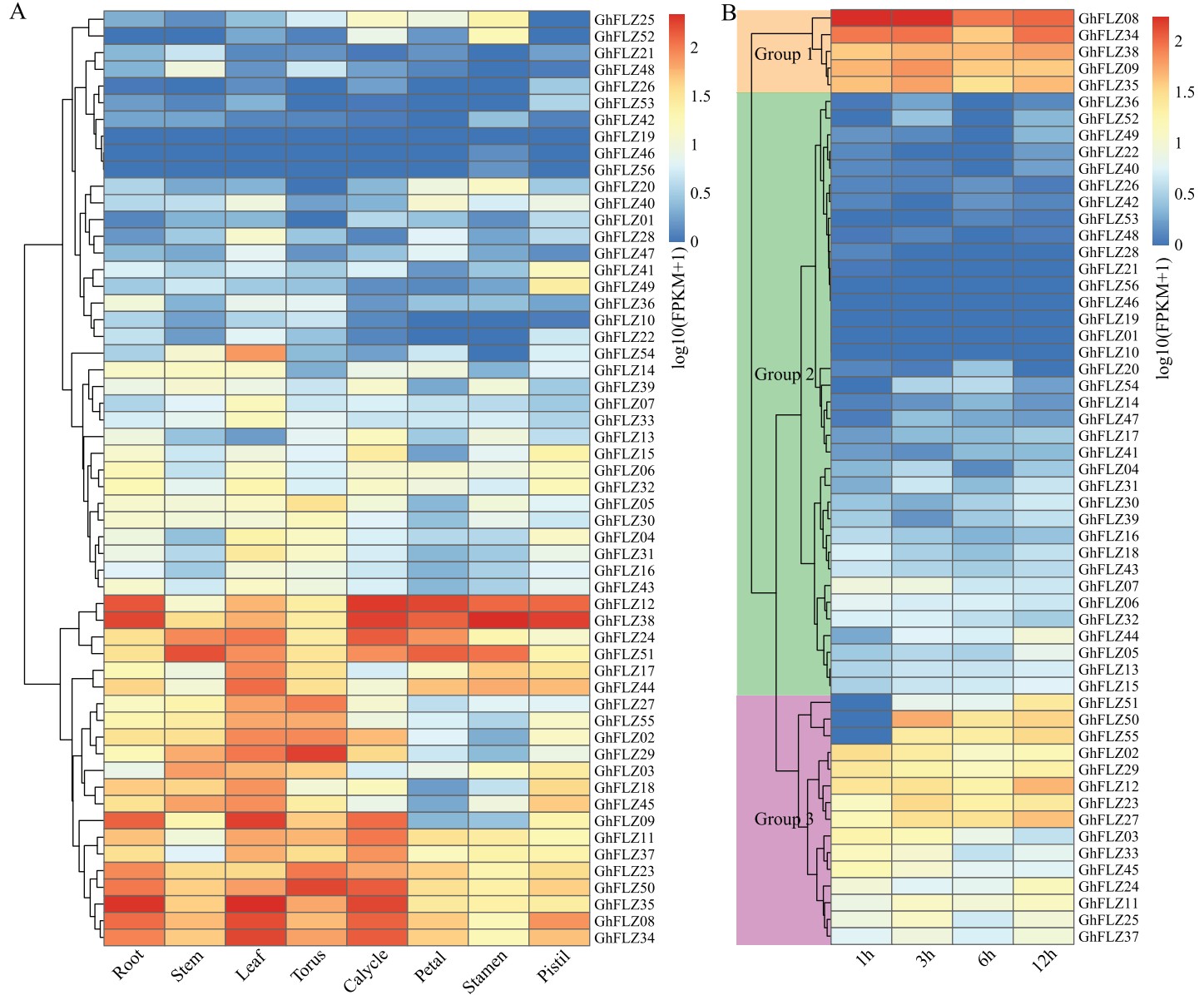

**Figure 6 Expression analysis of *G. hirsutum* FLZ gene.** (A) Tissue-specific expression analysis of FLZ genes from *G. hirsutum* based on transcriptome data. (B) Expression analysis of the FLZ genes from *G. hirsutum* under cold stress based on transcriptome data.

expressed (Fig. 6A). Three GhFLZ proteins (*GhFLZ19, GhFLZ46* and *GhFLZ56*) were expressed at low levels in all eight tissues, and the expression levels of these three proteins were not much different. Most of the genes were mainly expressed in the roots, stems, leaves, receptacles and calyx. This result indicated that except for *GhFLZ19, GhFLZ46* and *GhFLZ56*, the *G. hirsutum* FLZ gene family members had strong tissue expression specificity and contained more complex functions. The expression pattern of each GhFLZ gene was shown to be tissue specific in this study, but this specificity may be related to more refined tissues than those examined here.

The expression analysis of FLZ family genes in *G. hirsutum* under low-temperature stress showed that the expression patterns of all *G. hirsutum* FLZ family genes can be divided into three categories (Fig. 6B). After low-temperature stress, only 12 genes (*GhFLZ03, GhFLZ07, GhFLZ11, GhFLZ24, GhFLZ25, GhFLZ33, GhFLZ37, GhFLZ44, GhFLZ45, GhFLZ50, GhFLZ51* and *GhFLZ55*) showed evident changes in expression, while the remaining genes did not show visible changes. The expression of these 12 genes was induced by low-temperature stress; thus, these 12 genes may play corresponding roles in the response of *G. hirsutum* to low-temperature stress. Three genes with obvious expression level changes (*GhFLZ50, GhFLZ51* and *GhFLZ55*) were significantly upregulated and reached maximum levels at 12 h. However, since RNA-seq was only performed on samples collected under low-temperature stress up to 12 h, we speculate that the expression of these genes may have continued to increase after 12 h of low-temperature stress. These results suggest that *G. hirsutum* FLZ family genes may play a role in the response of *G. hirsutum* to low-temperature stress.

## qRT–PCR

The expression pattern of a gene is usually related to its function. Previous studies have shown that FLZ genes play an important role in plant responses to cold stress (*Jamsheer et al., 2018a*, *2018b*; *He et al., 2001*). According to the transcriptome expression profile, we speculated that 12 genes may be involved in the stress response of *G. hirsutum* to low temperature. To understand the expression patterns of these 12 GhFLZ genes in *G. hirsutum* in response to low-temperature stress, we evaluated their expression patterns in low-temperature-tolerant and low-temperature-sensitive materials by qRT–PCR. Compared with before low-temperature treatment, the expression levels of all but 3 GhFLZ genes (*GhFLZ37, GhFLZ50* and *GhFLZ51*) were significantly different at different time points after low-temperature stress (Fig. 7), indicating that these genes may be involved in cotton's defense against low-temperature stress. Among them, four genes (*GhFLZ11, GhFLZ25, GhFLZ44* and *GhFLZ55*) were also significantly different between the resistant and sensitive materials at the same time points. In summary, these four genes may all play a role in the response of *G. hirsutum* to low-temperature stress rather than having naturally high expression during the growth and development of cotton at this stage.

Due to gene expression levels being different in different tissues, the molecular biological functions of a gene product in different tissues are also different. To this end, we selected rhizomes and leaves under low-temperature stress for 24 h to further verify the tissue expression patterns of four genes (*GhFLZ11, GhFLZ25, GhFLZ44* and *GhFLZ55*) in these two materials (Fig. 8). There was no significant difference in *GhFLZ25* expression in any tissue of the two materials. The expression of *GhFLZ11* in the leaves of the 86-1 cultivar was 1.5-fold greater than that in the Lumian 2 cultivar. The expression levels of *GhFLZ44* and *GhFLZ55* in the leaves of the 86-1 cultivar were three-fold greater than those in the leaves of the Lumian 2 cultivar, and there were no significant differences in the expression of these genes in other tissues. The expression levels of *GhFLZ11* and *GhFLZ44* in leaves were significantly higher than those in roots and stems, and the expression levels in stems were the lowest in both low-temperature-resistant materials and low-

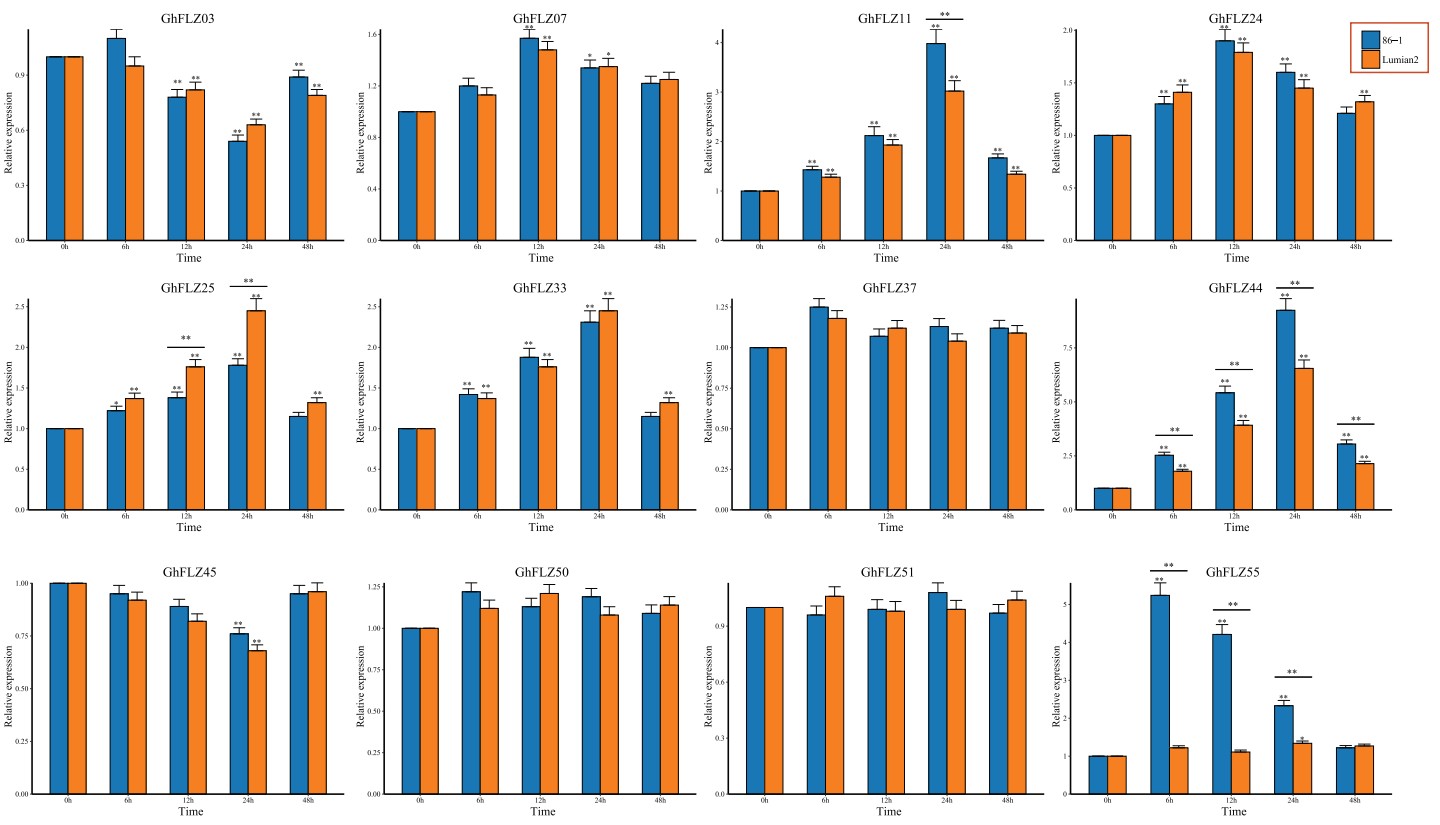

**Figure 7 Expression analysis of the FLZ gene in 86-1 and Lumian 2 under low temperature.** Error bars represent the average of three replicates ± SD. The difference from the control group is statistically significant, $^*P < 0.05$; $^{**}P < 0.01$.

temperature-sensitive materials. The expression levels of *GhFLZ25* in roots and leaves were higher than that in stems. The expression level of *GhFLZ55* in leaves was the highest in low-temperature-tolerant materials, but its expression levels in roots, stems and leaves were almost the same in low-temperature-tolerant and low-temperature-sensitive materials, with no significant differences. These results indicate that *G. hirsutum* FLZ gene expression is likely upregulated in response to low-temperature stress and that the significant changes in *G. hirsutum* FLZ gene expression in leaves lead to the improvement of cotton cold tolerance. The results of this study show that the expression pattern of the *G. hirsutum* FLZ gene family is complex, and more in-depth research is needed in the future to analyze the important role of the *G. hirsutum* FLZ gene family in low-temperature stress. The results of this study also lay a foundation for us to further verify the function and molecular mechanism of GhFLZ proteins and analyze the molecular basis of their role in low-temperature tolerance.

## DISCUSSION

Most of the Earth's land area (64%) has a minimum average temperature below 0 °C, but many crops, such as rice, wheat, soybean and cotton, lack the ability to adapt to low temperatures, so they can only live in tropical and subtropical regions (*Shi et al., 2020*; *Yan et al., 2020*; *Tchagang et al., 2017*; *Chen et al., 2020b*; *Akhtar & Farooq, 2019*). Therefore,

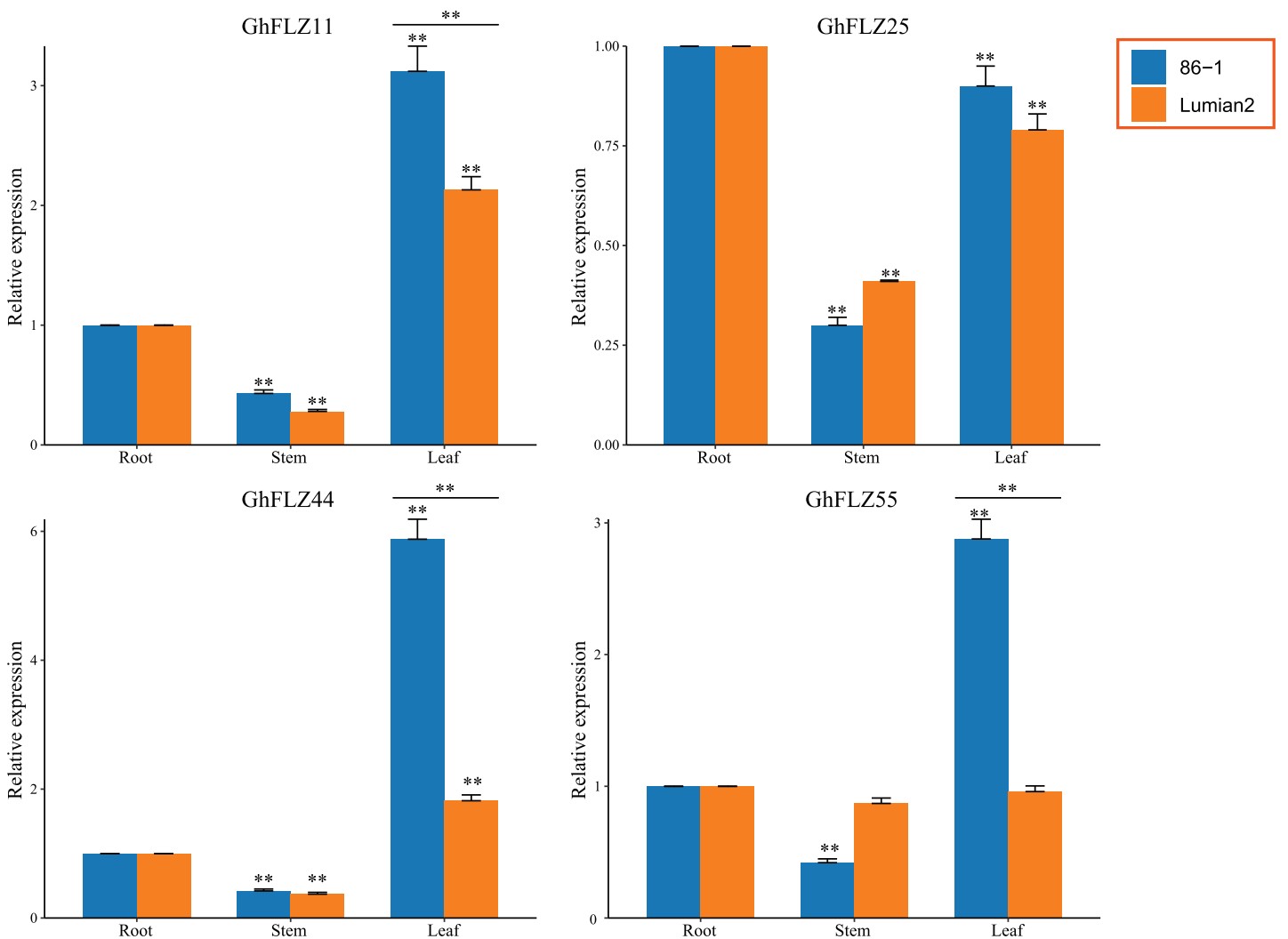

**Figure 8 Expression analysis of *GhFLZ11*, *GhFLZ25*, *GhFLZ44* and *GhFLZ55* in 86-1 and Lumian2 under low temperature conditions.** Error bars represent the average of three replicates ± SD. The difference from the control group is statistically significant, $^{**}P < 0.01$.

low temperature severely affects the environmental factors of plant growth and development, restricts the geographical distribution of plants, and affects crop yield (*Yan et al., 2020*; *Tchagang et al., 2017*; *Akhtar & Farooq, 2019*). Plants have evolved a series of mechanisms that enable them to adapt to cold stress at the physiological and molecular levels (*Tchagang et al., 2017*). In the past two decades, much work has been devoted to unearthing the key elements of plant cold tolerance and dissecting their regulatory mechanisms. The mining of low-temperature tolerance candidate genes plays an important role in this process (*Shim et al., 2019*). The FLZ gene family plays an important role in plant growth and development and stress resistance (*Jamsheer et al., 2018a*, *Hou et al., 2013*; *He et al., 2001*). In recent years, the identification and functional studies of plant FLZ family genes have attracted increasing attention. To date, most of the information about FLZ family genes has come from studies in *Arabidopsis*, rice and maize,
while a systematic analysis of the FLZ gene family in cotton has not been reported (*Hou et al., 2013*; *He et al., 2001*; *Laxmi, 2014*). In this study, we performed a systematic identification and evolutionary analysis of FLZs in cotton and provided the gene and protein properties, phylogenetic relationships, and gene expression patterns of 56 GhFLZ proteins in different tissues as well as their subcellular localization in *G. hirsutum*.

FLZ family genes are widely present in plant genomes, and earlier studies identified FLZ family genes from plants, such as maize and *Arabidopsis* (*Hou et al., 2013*; *He et al., 2001*; *Chen et al., 2021*). Compared with the FLZ family genes of these plants, the GhFLZ family genes have both uniform characteristics and several species specificities. First, in terms of the sequence length of the FLZ protein, the number of amino acid residues in the rice FLZ protein (OsFLZ) does not exceed 400, which is basically the same as the length as that of the GhFLZ protein. Second, all of the proteins encoded by the plant FLZ genes found thus far contain only a single FLZ domain and do not contain a transmembrane domain or signal peptide sequence, which are both present in the OsFLZ protein (Fig. S1). Both proteins contain the conserved CX2CX3LX4DX3YX5FCSX2CR motif and exhibit α-β-α secondary topological features (Fig. 4). Intron–exon structural analysis, which is commonly used in the study of gene evolution, also showed that GhFLZ, similar to other plants, only contains a single intron (*Chen et al., 2021*). Consistently, the collinearity analysis showed a large number of collinear blocks among the FLZ genes of *G. hirsutum*, rice and *Arabidopsis thaliana*, and there were 27 pairs of *G. hirsutum*, rice and *Arabidopsis* collinear genes (Fig. 3A). In addition, the FLZ family gene sequences in the A subgenome of *G. hirsutum* were collinear with those of *G. arboreum* and *G. barbadense*, and the FLZ gene sequences in the D subgenome were collinear with one of the sequences in *G. raimondii* and *G. barbadense* (Fig. 3B). These results suggest that the upland FLZ was relatively conserved during the evolution of cotton. Based on these results, we speculate that the FLZ gene of dicotyledonous plants may have evolved from a common ancestor and diverged earlier in evolution than the FLZ gene of monocotyledonous plants.

Transcription factors (TFs) regulate plant growth, development and stress resistance by regulating the expression of genes mainly involved in responses to hormones and environmental factors, as well as cell differentiation and organ development (*Amorim et al., 2017*). Our analysis of the *cis*-elements in the promoter region of the *G. hirsutum* FLZ genes showed that MeJA- and ABA-related hormone-responsive elements are the most abundant in *G. hirsutum*, suggesting that the *G. hirsutum* FLZ family may exert its biological functions mainly through the MeJA and ABA pathways (Fig. 6, Fig. S2). Studies have shown that *miR1320* positively regulates cold tolerance in rice by inhibiting the expression of *OsERF096* and relieving the inhibition of *OsERF096* by the JA-mediated CBF signaling pathway (*Sun et al., 2022*). Moreover, in watermelon, exogenous application of MeJA can significantly improve cold tolerance (*Li et al., 2021*). During periods of low temperature, the endogenous ABA content in plants increases, and the accumulated ABA functions to regulate genes that are regulated by low temperature, thereby improving the cold resistance of plants (*Chen et al., 2018b*; *Zhang et al., 2019*). These results suggest that the FLZ gene family may also be involved in the response of *G. hirsutum* to low temperature through the MeJA and ABA pathways.

Through expression pattern analysis, the expression levels of four genes (*GhFLZ11, GhFLZ25, GhFLZ44* and *GhFLZ55*) were shown to be changed significantly at different time points of low-temperature stress, and there were also significant differences in expression in different tissues. The expression of these four genes in leaves was generally higher than that in stems and leaves (Fig. 8). What is more interesting is that the expression levels of these four genes were significantly different in roots, stems and leaves before and after different stresses, but the expression in stems and roots was not as high as that in leaves. Previous studies have shown that ABA can activate SnRK1 by inhibiting ABI1 and PP2CA, while the FLZ protein interacts with the SnRK1 kinase subunit to generate a fast-folding SnRK1 protein (*Jamsheer et al., 2018a*; *Hou et al., 2013*). There is a strong interaction between the FLZ protein and SnRK1 in *Arabidopsis* and maize (*Kaur Dhaliwal et al., 2021*; *Jamsheer et al., 2019*). This further indicates that *G. hirsutum* FLZ gene expression is likely affected by low-temperature stress, and the significant change in its expression in leaves causes the expression of MeJA and ABA-related genes to regulate the opening and closing of leaf stomata to improve the heat resistance of cotton.

In summary, our tissue and low-temperature stress expression profiles of cotton, as well as our preliminary analysis of the *G. hirsutum* FLZ gene family using qRT–PCR, suggest that members of this family play an important role in *G. hirsutum* under low-temperature stress. Moreover, the expression of *G. hirsutum* FLZ genes is likely to be significantly changed in leaves after the plant is subjected to adversity stress to improve the adaptation of cotton to cold stress. In conclusion, these results lay the foundation for us to further verify the function of cotton FLZ proteins and analyze their molecular mechanism of low-temperature tolerance.

## CONCLUSION

In this study, genome-wide identification of cotton FLZ genes was performed, and 56 FLZ genes were included in the *G. hirsutum* genome, significantly more than those in *Arabidopsis* and rice. The results of phylogenetic tree and collinearity analysis indicated that GhFLZ gene family members can be divided into five large subgroups, which are relatively conserved in the evolution of cotton. Through promoter and expression analysis, *GhFLZ11, GhFLZ25, GhFLZ44* and *GhFLZ55* were found to be important regulatory genes in the response of *G. hirsutum* to low-temperature stress. This study is the first to systematically analyze the cotton FLZ gene family and provides a new understanding of cotton resistance to low-temperature stress, which lays a foundation for the in-depth functional analysis and breeding application of GhFLZ.

## ACKNOWLEDGEMENTS

We thank the "Tianshan" Innovation team program of the Xinjiang Uygur Autonomous Region (2021D14007) for providing superior cotton varieties and the Collaborative Improvement of Cotton Yield and Quality for mechanical harvesting.

### Funding

This work was supported by the Major Science and Technology Project of Xinjiang Uygur Autonomous Region (2021A02001-4). The funders had no role in study design, data collection and analysis, decision to publish, or preparation of the manuscript.

### Grant Disclosures

The following grant information was disclosed by the authors:
Major Science and Technology Project of Xinjiang Uygur Autonomous Region: 2021A02001-4.

### Competing Interests

Junduo Wang, Yajun Liang, Juyun Zheng, Zhaolong Gong, Xueyuan Li are employees of the Cash Crops Research Institute of Xinjiang Academy of Agricultural Science, China. Zhiqiang Li, Guohui Zhou and Yuhui Xu are employed by Adsen Biotechnology Co., Ltd., China. The authors declare that they have no competing interests.

### Author Contributions

- JunDuo Wang conceived and designed the experiments, performed the experiments, analyzed the data, prepared figures and/or tables, authored or reviewed drafts of the article, and approved the final draft.
- Zhiqiang Li conceived and designed the experiments, performed the experiments, analyzed the data, prepared figures and/or tables, authored or reviewed drafts of the article, and approved the final draft.
- Yajun Liang analyzed the data, prepared figures and/or tables, authored or reviewed drafts of the article, and approved the final draft.
- Juyun Zheng analyzed the data, prepared figures and/or tables, authored or reviewed drafts of the article, and approved the final draft.
- Zhaolong Gong analyzed the data, prepared figures and/or tables, authored or reviewed drafts of the article, and approved the final draft.
- Guohui Zhou analyzed the data, prepared figures and/or tables, authored or reviewed drafts of the article, and approved the final draft.
- Yuhui Xu conceived and designed the experiments, analyzed the data, prepared figures and/or tables, authored or reviewed drafts of the article, and approved the final draft.
- Xueyuan Li conceived and designed the experiments, analyzed the data, prepared figures and/or tables, authored or reviewed drafts of the article, and approved the final draft.

### Data Availability

The sequence data is available at NCBI SRA: PRJNA248163. The PCR data are available in the Supplemental File.

## Supplemental Information

Supplemental information for this article can be found online at http://dx.doi.org/10.7717/peerj.14690#supplemental-information.

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
