# Peer review of "Genome-wide identification and expression reveal the involvement of the FCS-like zinc finger (FLZ) gene family in Gossypium hirsutum at low temperature"

_PeerJ, doi:10.7717/peerj.14690_

## Round 0.1 · original submission · Major Revisions

Please revise the manuscript as per the comments of the reviewers and resubmit for consideration

Regards
Mohan Lal

Reviewer 1 ·

Basic reporting

no comments

Experimental design

experiment was not planned properly as three leaf stage is two early to predict the effect of chilling and low temp on fibre yield and quality eventaully harvest a information on gene expression.

Validity of the findings

No information was provided about effect of chilling treatment on both genotypes

Additional comments

Introduction not justifying the experiment
Actual lab information not proper experiment (maximum work is in-silico)

Annotated reviews are not available for download in order to protect the identity of reviewers who chose to remain anonymous.

Reviewer 2 ·

Basic reporting

The manuscript shows the identification and expression analyses of FLZ genes in Gossypium hirsutum with the aim of detecting key genes related to low-temperature stress. The manuscript shows interesting data about the evolution and structure of FLZ genes, as well as expression patterns of these genes in response to cold stress. The introduction and background are well supported. The structure of the article is acceptable, however, some results presented are not properly described so it is difficult to fully understand the information presented.

For a better understanding of the data presented, the authors should improve the results description and legends of the figures. The legend must contain all the necessary to understand the information presented.

Figure 6: The meaning of color scale values must be included in the legend, what represents these values?

Figure 8: It suggests restructuring this figure, each graph needs to include the information on the expression of each gene in both genotypes (resistant and susceptible) and the tissues evaluated, as the figure 7. In this way, it will be possible to determine the possible role of these genes in response to cold stress.

Experimental design

The authors show interesting information about the function of FLZ genes in response to cold stress, however, it is necessary to resolve several points in the methods described.

Lines 91-95: This part is not clear. Model of climate incubator? What means "the light was 8"? Which were the two treatments? In addition, there is no information about how many biological replicates for low-temperature stress experiments were performed. Please, describe in detail how many biological replicates were used? and how many plants were used for biological replicates?
Lines 125-126: It is necessary to describe in more detail how was performed the low-temperature stress experiment or indicate a reference. In addition, it suggests including the SRX number of RNA-seq samples used in this analysis. The PRJNA248163 number refers to the genome G. hirsutum project, which hasn't been cited in this manuscript (Zhang, T., Hu, Y., Jiang, W. et al. Sequencing of allotetraploid cotton (Gossypium hirsutum L. acc. TM-1) provides a resource for fiber improvement. Nat Biotechnol 33, 531–537 (2015). https://doi.org/10.1038/nbt.320).
Lines 127-131: It is necessary to describe in more detail how was carried out the filtering and quality control of reads.

Validity of the findings

It is recommended to double-check the description of the results shown because the manuscript has several inconsistencies.

The authors mention in the abstract that the G. hirsutum genome contains 59 FLZ genes, however, in the results section (line 150) they mention that 56 were identified, and then in line 153, they mention 69. Please review and correct these inconsistencies.

Lines 244-249: The authors claim that the genes have tissue-specific expression, however, Figure 6A shows that most genes are expressed in all tissues, except for some examples such as those GhFLZ48 and GhFLZ28 genes, among others. This section needs to be clarified.

Lines 250-254: The authors claim that the expression patterns of FLZ genes in response to low temperatures can be divided into three categories, however, the authors do not describe these three categories. Furthermore, the authors say that some genes significantly changed in expression (Figure 6B), however, there is no statistical evidence to support this claim. In addition, it is not clear where these data come from. The authors should describe in detail the origin and analysis of these data in the materials and methods section. Additionally, the description of this section must be improved.
What happens with GhFLZ08, GhFLZ34, GhFLZ38, GhFLZ09, and GhFLZ35 genes? These have the highest expression during low-temperature stress.

It is suggested to restructure figure 8 to be able to compare the expression levels of the genes (GhFLZ11, GhFLZ25, GhFLZ44, GhFLZ55) between the resistant and susceptible genotypes. This will allow analyzing the genotype effect on the expression of these genes in response to cold stress, which will help to support the conclusions made.

In general, the section discussing patterns of expression is weak. Considering that this part provides more information on the FLZ genes' participation in the cold stress response, the authors should improve and deepen the discussion of these results.

Lines 249-251: The authors claim that significant changes in the expression of FLZ genes in leaves "causes the expression of MeJA and ABA-related genes to regulate the opening and closing of leaf stomata..." What evidence is there that can support the possible involvement of FLZ genes in the opening and closing of stomata in response to cold? Please include information that supports it.

Additional comments

It is recommended to check the manuscript to correct typographical errors…

---

## Round 0.2 · Major Revisions

Dear Authors

Reviewer 1 has raised serious issues in the manuscript that were not resolved in revision. Hence, in its current form article can not be considered for publication however if you feel that you can revise it so, I will consider it for further review. Please revise according to the earlier comments of the reviewers.

Please note: This is the final opportunity to defend your manuscript.

Reviewer 1 ·

Basic reporting

.

Experimental design

.

Validity of the findings

.

Additional comments

.

Reviewer 2 ·

Basic reporting

The present manuscript version was improved significantly compared to the previous one. The authors responded to the observations and suggestions made.
It is suggested to edit the sentence (lines 312-316): "...were obviously changed in expression, and the remaining genes also did not change obviously." change to "...showed evident changes in expression during cold stress, while the remaining genes did not show visible changes."

Experimental design

no comment

Validity of the findings

no comment

---

## Round 0.3 · accepted · Accept

All the queries raised by the reviewers have been successfully resolved during the revision

Reviewer 1 ·

Basic reporting

Article revised as per suggestions/queries, may be accepted

Experimental design

.

Validity of the findings

.

Additional comments

.